# A 3D-printed condom intrauterine balloon tamponade: Design, prototyping, and technical validation

Davide Piaggio[1]*, Scott Hyland[1], Alessia Maccaro[1], Ernesto Iadanza[2], Leandro Pecchia[1,3]

**1** School of Engineering, University of Warwick, Coventry, United Kingdom, **2** Department of Medical Biotechnologies, University of Siena, Siena, Italy, **3** School of Engineering, Campus Biomedico of Rome, Rome, Italy

* davide.piaggio@warwick.ac.uk

**Data Availability Statement:** The datasets used and/or analysed during the current study are available as a Supplementary Material.

## Abstract

Post-partum haemorrhage is among the main causes of (preventable) mortality for women in low-resource settings (LRSs), where, in 2017, the mortality ratio was 462 out of every 100 000 live births, over 10 times higher than for high-resource settings. There are different treatments available for post-partum haemorrhage. The intrauterine balloon tamponade is a medical device that proved to be a simple and cost-effective approach. Currently, there are several balloon tamponades available, with different design and working principles. However, all these devices were designed for high-resource settings, presenting several aspects that could be inappropriate for many lower-income countries. This paper presents the results of a preclinical study aiming at informing the design, prototyping and validation of a 3D-printed intrauterine balloon tamponade concept, contributing towards the United Nation's Sustainable Development Goal 3: Good health and Well-being. Frugal engineering concepts and contextualised design techniques were applied throughout, to define the design requirements and specifications. The performance of the final prototype was validated against the requirements of the UK National Health System (NHS) technical guidelines and relevant literature, measuring the water leak and pressure drop over time, both open air and in a approximate uterus model. The resulting prototype is made up of six components, some of which are easy to retrieve, namely a water bottle, a silicone tube and an ordinary condom, while others can be manufactured locally using 3D printers, namely a modified bottle cap, a flow stopper and a valve for holding the condom in place. Validation testing bore promising results with no water or pressure leak open air, and minimal leaks in the approximate uterus model. This demonstrates that the 3D printed condom-based intra-uterine balloon tamponade is performing well against the requirements and, when compared to the state of the art, it could be a more appropriate and more resilient solution to low-resource settings, as it bypasses the challenges in the supply of consumables and presents a greener option based on circular economy.

**Funding:** DP and LP received support from the University of Warwick with two Warwick Impact Found grants supported by the EPSRC Impact Accelerator Award (EP/K503848/1 and EP/R511808/1). AM's Fellowship is supported by the WIRL COFUND – Marie Sklodowska Curie Actions, Institute of Advanced Study, University of Warwick (UK).

**Competing interests:** The authors have declared that no competing interests exist.

## Introduction

Although the number of women dying during pregnancy and childbirth has decreased by 43% since the 1990 [1], there is still a growing gap between LRSs and higher resource ones. In fact, women living in a low- and middle-income (LMIC) have approximately a 33 times higher chance of dying from complications during pregnancy and childbirth in respect to those living in high-income countries (HICs) [1]. The World Health Organisation (WHO) reported that in 2017 the maternal mortality ratio is 462 per 100,000 live births for LMICs (11 per 100,000 in HIC) [2]. The WHO also affirmed that LRSs accounted for 94% of mostly preventable deaths during pregnancy and childbirth in 2017 [3].

One of the most common causes of maternal death worldwide is postpartum haemorrhage [4], affecting 4% of all pregnancies and accounting for 25% of maternal mortalities worldwide (a rate that has been unchanged since 1992) [5]. Postpartum haemorrhage is defined as heavy bleeding after birth and it can be classed as "primary" if there is a 500-ml blood loss (>2 litres in case of severe haemorrhage) within the first 24 hours after birth, and as "secondary" if there is abnormal or heavy vaginal bleeding between 24 hours and 12 weeks after birth [6]. The most common causes of postpartum haemorrhage are uterine atony, abnormal placentation as well as retained placenta, genital tract lacerations and rupture, and coagulopathy [7, 8]. As per the WHO [9] and the International Federation of Gynecology and Obstetrics (FIGO) guidelines [10], first-line treatment of postpartum haemorrhage includes the use of uterotonics (e.g., oxytocin) and of tranexamic acid. If women do not respond to such treatments or they are not available, uterine massage and intrauterine balloon tamponades are effective non-surgical alternatives, always according to WHO and FIGO. If all the above methods fail, the use of uterine artery embolization is recommended for treating postpartum haemorrhage due to uterine atony. Some measures are also recommended as temporary solutions, such as bimanual uterine compression, external aortic compression, and non-pneumatic anti-shock garments. Otherwise, also surgical intervention is one possible solution.

Relative to this, the UN is aiming to "reduce the global maternal mortality ratio to less than 70 per 100,000 live births" by 2030 [11]. In light of this goal, with regards to intrauterine balloon tamponades, many solutions of different shapes, sizes and materials were released, including: the Bakri tamponade balloon catheter, the BT-Cath, the ebb tamponade system, the Every Second Matters uterine balloon tamponade, the Alves handmade intrauterine balloon [12], the Kyoto balloon system [4] a free-flow pressure controlled uterine balloon [13]. In addition to these, other devices designed for other purposes have been used for treating postpartum haemorrhage, including Sengstaken-Blakemore tube, Foley catheters, condom catheters and surgical gloves [1].

Although not initially conceived for this purpose, condom catheter balloons prove to be a low-technology, non-invasive, safe, effective and easy-to-use solution for the treatment of postpartum haemorrhage. Above all in LRSs, where there is insufficient training, low numbers of experienced personnel and a lack of preassembled intrauterine balloon tamponades [7, 14, 15]. A recent metanalysis [16] and other contextual studies on the use of such solutions in African countries (e.g., Ghana, Kenya, Egypt, etc.) confirmed a high success rate, ranging from 88.6% to 96% [5, 7, 15]. One study related to Sierra Leone and Kenya [17] also reported a significantly lower mean blood loss, lower occurrence of blood transfusions, lower intensive care unit admission rates and lower occurrence of infectious morbidities. The same authors reported lower success rates in cases of Caesarean sections and advanced maternal age. Nonetheless, Suarez et Al. [16] affirmed that further studies are required, as evidence on any uterine balloon tamponade efficacy and effectiveness from randomized and non-randomized studies is contrasting. Also Anger et Al. [18] claimed that intrauterine balloon tamponades increase the

chances of postpartum haemorrhage related to surgery and death. This paper wreaked havoc in the scientific community and received commentaries by two specialists, S. Matsubara et Al. [19] and by Weeks [20]. Overall, the authors agreed regarding the fact that the sample size was too small, a new technology had been introduced without proper training, no techniques to mitigate the prolapse of the balloon were used, and that there were system issues at play (e.g., understaffing, referral systems, infrastructures, consumables, training, corruption, etc.). In this scenario, Candidori et al. [21, 22] have taken advantage of the condom balloon tamponade design and proposed BAMBI, a similar device with the only difference of using a 3D-printed connector for the condom rather than sutures.

Here in this study, a unique 3D-printed intrauterine balloon tamponade is conceptualized, constructed, and verified, based on the above described findings and a circular economy approach. To guide the design process with a rigorous regulatory approach, the European Medical Device Regulation (EU) 2017/745 on medical devices (MDR 2017/745) was followed, together with its European Medical Device Nomenclature (EMDN) as per Article 26. In particular, our device was classified as part of the devices called "postpartum intrauterine catheters" (U100102). Although dealing with a design concept project, the relevant first regulatory steps were followed to lay solid basis for future regulatory approval processes. In particular, the intended use was specified as follows: the 3D-printed condom-based intrauterine balloon tamponade is meant to be used by doctors, nurses or trained lay users, to provide means of temporary control or reduction of postpartum hemorrhage in patients for whom the first line of treatment of such condition (e.g., uterotonics) does not work. The system can be used in any healthcare setting, ambulances, or at the patient's home, should there be no other option available.

The device, according to the classification rules presented in Annex VIII of MDR 2017/745, falls into Class IIa. Specifically, Rule 5 of Annex VIII reads "*other than surgically invasive devices, which are not intended for connection to an active device or which are intended for connection to a class I active device are classified as [. . .] class IIa if they are intended for short-term use [i.e., from 60 minutes to 30 days, except if they are used in the oral cavity as far as the pharynx, in an ear canal up to the ear drum or in the nasal cavity, in which case they are classified as class I*".

Together with MDR 2017/745 requirements, the available technical guidance from the UK NHS was followed to ensure the safety and efficiency of this medical device (MD).

## Materials and methods

### Contextualised design

Several field studies were run over the past 6 years in four sub-Saharan African (SSA) countries, namely Benin, Ethiopia, Uganda and South Africa [23, 24], using a combination of quantitative engineering methods and qualitative ethnographic and bioethical ones. During these field-experiences, a SWOT analysis was performed, healthcare structures were inspected and assessed, healthcare personnel, biomedical engineers and technicians, and sociologists were interviewed to inform the design of different MDs, including this intrauterine balloon tamponade.

### CAD design and 3D printing

The 3D printed parts were designed on Autodesk Fusion360 [25], sliced on Cura [26] and printed on a Creality Ender3 3D printer using black polylactic acid (PLA), as it is a very versatile material and is biocompatible. The parts were: 1) A valve 2) A flow-stopper 3) A modified bottle cap 4) A uterus model. Components 1–3 are the parts of the final system that could be

eventually recycled and protocycled with the addition of up to 50% virgin PLA pellets into new PLA filament.

**Valve.** Four different designs, namely two based on a snap-fit connection and two on a threaded connection, were created, printed, and tested. The valves were printed with 0.16 mm layer height and 15% infill, 200˚C nozzle temperature and 40˚C bed temperature. In particular, the threaded valves were printed so that their thread axes would be perpendicular to the print bed for best results. The selected threads were: M11x0.75 (finer) and M11x1.5 (coarser). All the valves were designed so that no overhang support would be needed during the 3D printing phase.

**Flow stopper.** The flow-stopper is a type of globe valve that has two parts: a base through which a silicon tube runs and a screw that operates on the tube to restrict its lumen until there is no space for water to flow back.

**Modified bottle cap.** The bottle cap was designed following the PCO1881 [27] standard for threads compatible with common soda bottles. It was modified with a truncated cone-like additional structure to act as a channel for the silicon tube. Moreover, the cap was designed with two seals, namely a v-seal and a plug seal [28].

**Uterus model.** Mollazadeh-Moghaddam et al. [29] proposed a set of models to mimics to shapes of uteri, based on literature data [30–33]. Specifically, they proposed three models, featuring 100, 250, and 500 mL volumes and cervix openings increasing in size, i.e., 1.5, 2, 2.5 cm of diameter. Similarly, a post-pregnancy uterus model was designed and 3D printed with PLA, so that the internal volume would be of 400 ml and the cervix opening would have a diameter of 3 cm, resembling the conditions of uteri of post-labour women. This model, consisting of two threaded parts, was printed with 0.16 mm layer height and 10% infill.

## Other parts and assembly of the system

Among the other parts, a common polyethylene terephthalate (PET) 500 mL bottle was used as a water reservoir, a standard latex condom was used as an inflatable balloon tamponade and a standard silicon tube (with an external diameter of 6 mm, 50 cm in length) was used for channelling water in the balloon. The assembly steps follow: 1) Insert the tube (6-mm outer diameter, 4-mm inner diameter) in both the bottle cap, the flow stopper and the valve; 2) Insert the condom on the male part of the valve and through the female part of the valve; 3) Screw the valve tight, paying attention not to excessively stretch the condom; 4) Fill the bottle with water; 5) Screw the bottle cap onto the bottle.

## Technical validation

The system was assembled for each valve type and two validation procedures were performed, following the methods that Mollazadeh-Moghaddam et al. presented in [29]: i) water leak measurements open-air and in uterus model, ii) pressure measurements in uterus model. Fig 1 shows the inflation of the system and its technical validation.

**Water leak.** For the water leak experiments, the condom was tested with two fill volumes, i.e., 400–500 mL and/or 1500–2000 mL of water, repeating the filling 3 times with a 500 mL bottle. The water leaked out of the system was weighed with a digital scale and recorded hourly for up to 6 hours, as this is the suggested duration of treatment [34]. The test results were used to improve the designs (e.g., moving from a finer to a coarser thread) and to compare all the valve prototypes. Eventually, the water leak of the best-performing valve was tested on two different prints, twice, in order to test two filling volumes, i.e., 400 and 1500 mL (see Fig 2 for experiment setup).

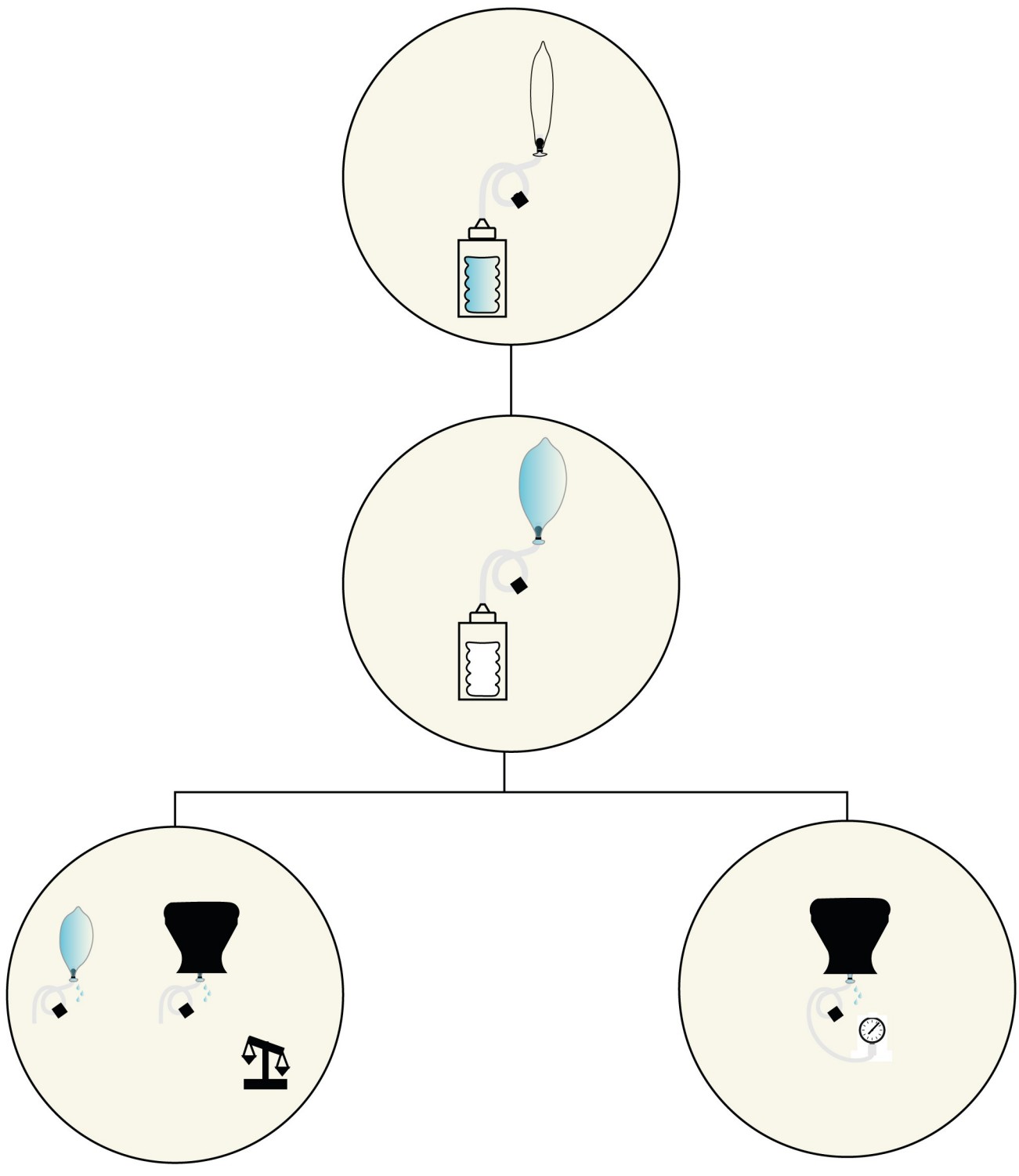

**Fig 1. The inflation of the system and the two tests, i.e., water leak tests and pressure drop tests, carried out in two conditions, i.e., open air and simulated working conditions.**

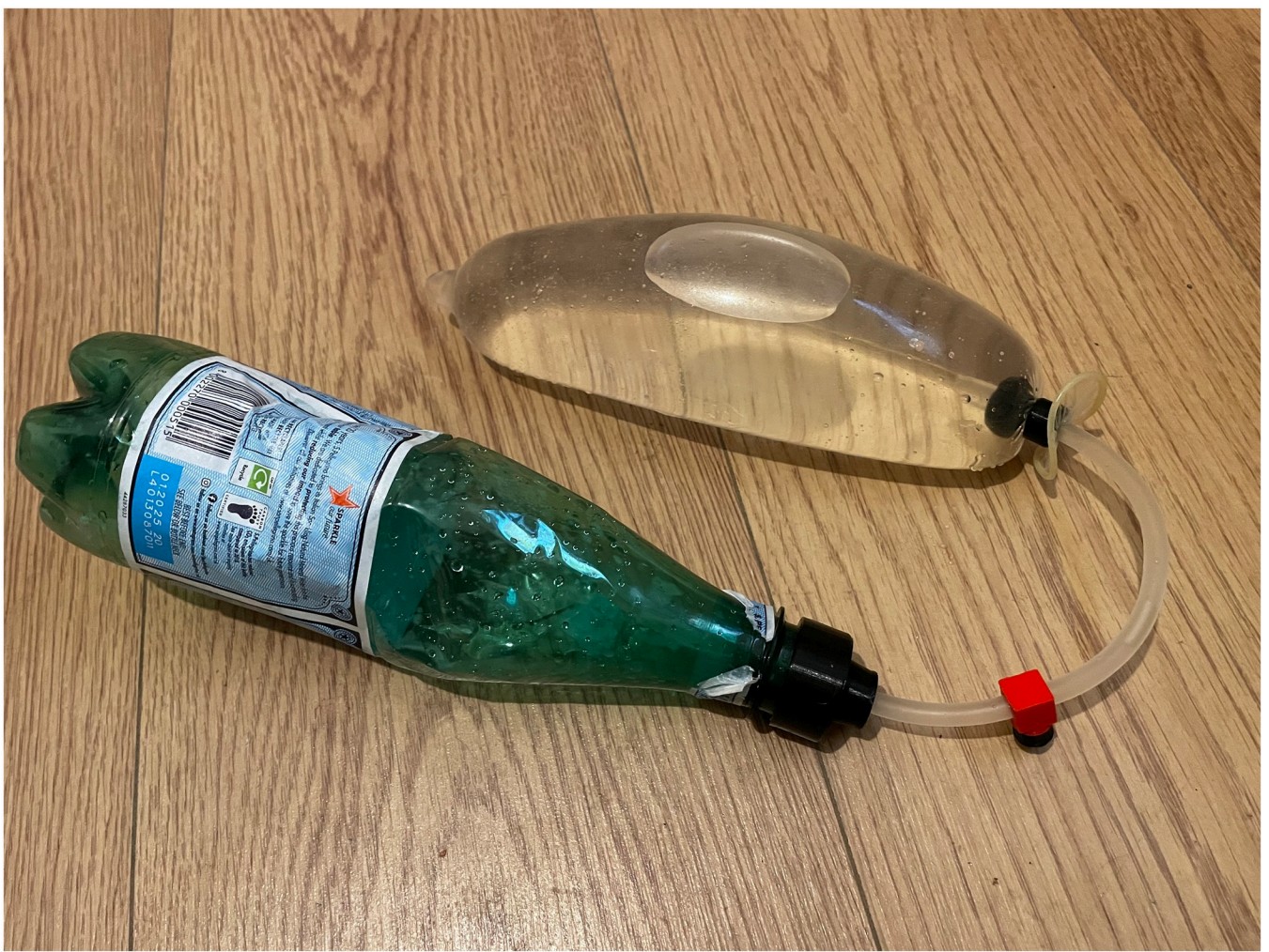

**Fig 2. The setup of the open-air experiments: The condom filled with water and the flow-stopper that prevents the water from flowing back through the tube.**

**Water leak in uterus model.** The uterine model was used to evaluate the system with the best valve. Two copies of the same prototype were tested in two positions, namely one in which the uterus model was suspended so that the "cervix" opening would point downwards, and gravity was acting on the balloon so that it pushed on the "cervix" (worst case scenario, 90˚) (position 1), and one in which the uterus model was laid so that the longitudinal axis would be almost parallel to the ground (14.7˚ estimated from the CAD) (position 2) (see Fig 3). The water leaked out of the system was weighed with a digital scale and recorded hourly for up to six hours.

**Pressure change in uterus model.** The same experiments for measuring water leaks with the system applied to the uterus model were then repeated for acquiring lumen pressures, hourly for up to three hours. The results from two experiments performed in the worst-case positioning of the uterus model (90˚) were then collected and averaged. For this purpose, an ad-hoc water pressure circuit was assembled using an Arduino kit and a 26PCCFB6G water pressure sensor by Honeywell. The water pressure circuit was validated using parts from a sphygmomanometer (i.e., the inflation bulb and the manometer) to increase the pressure in

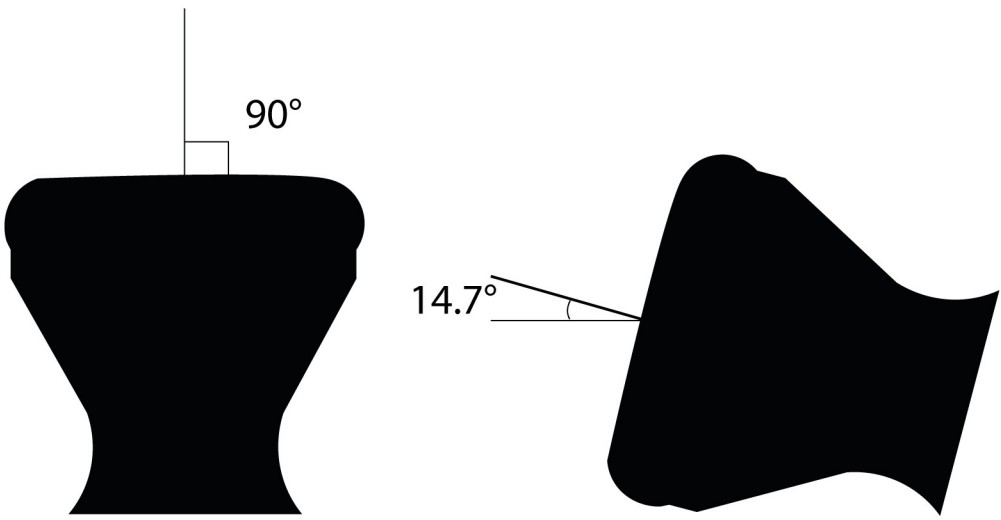

**Fig 3. The two positions of the uterus model during the water leak and pressure test.**

the tube and to compare the readings through a Bland-Altman plot, using the Matlab tool presented in [35].

## Comparison with other balloons

In order to allow for better comparison with other existing balloons, a proactive risk assessment process was carried out, using failure modes and effects analysis (FMEA) [36], for our device, the Bakri balloon, condom intrauterine balloons, and BAMBI. The FMEA allowed to highlight and compare the mechanical, biological, usability, cost, and clinical risks.

## Results

### Contextualised design

The design of our intrauterine balloon tamponade was informed by quantitative engineering and qualitative ethnographic and bioethical methods in field studies performed in SSA, the full results of which are reported elsewhere [23, 24, 37]. In summary, the field studies highlighted the chronic challenges typical of LRSs, including: the lack of spare parts, consumables, expertise, functional health technology management and maintenance, funds, and the presence of harsh environmental conditions (i.e., high temperatures, sand, dust, and humidity). Further considerations stemming from these are included in the Discussions section.

### CAD design and 3D printing

The principal function of the intrauterine balloon tamponade in treating postpartum haemorrhage is to inflate and provide pressure in the uterus, as shown in Fig 4. The prototype and constituent parts are presented in Figs 5–7, whereas Fig 8 shows the prototype of the uterus model. The 3D-printed parts totaled to roughly 10 grams of PLA (average market price currently at 20 dollars per kilogram), costing approximately 20 cents of US dollars. Taking into account an average price of 60 cents of dollar of a PTE 500-ml bottle, an average price of 10 cents of US dollars for the silicone tubing, and an average price of 15 cents of US dollars of a regular condom, the total system price approximately amounts to 1.05 US dollars. These costs would also be amortised and balanced by the fact that the 3D printed components could be

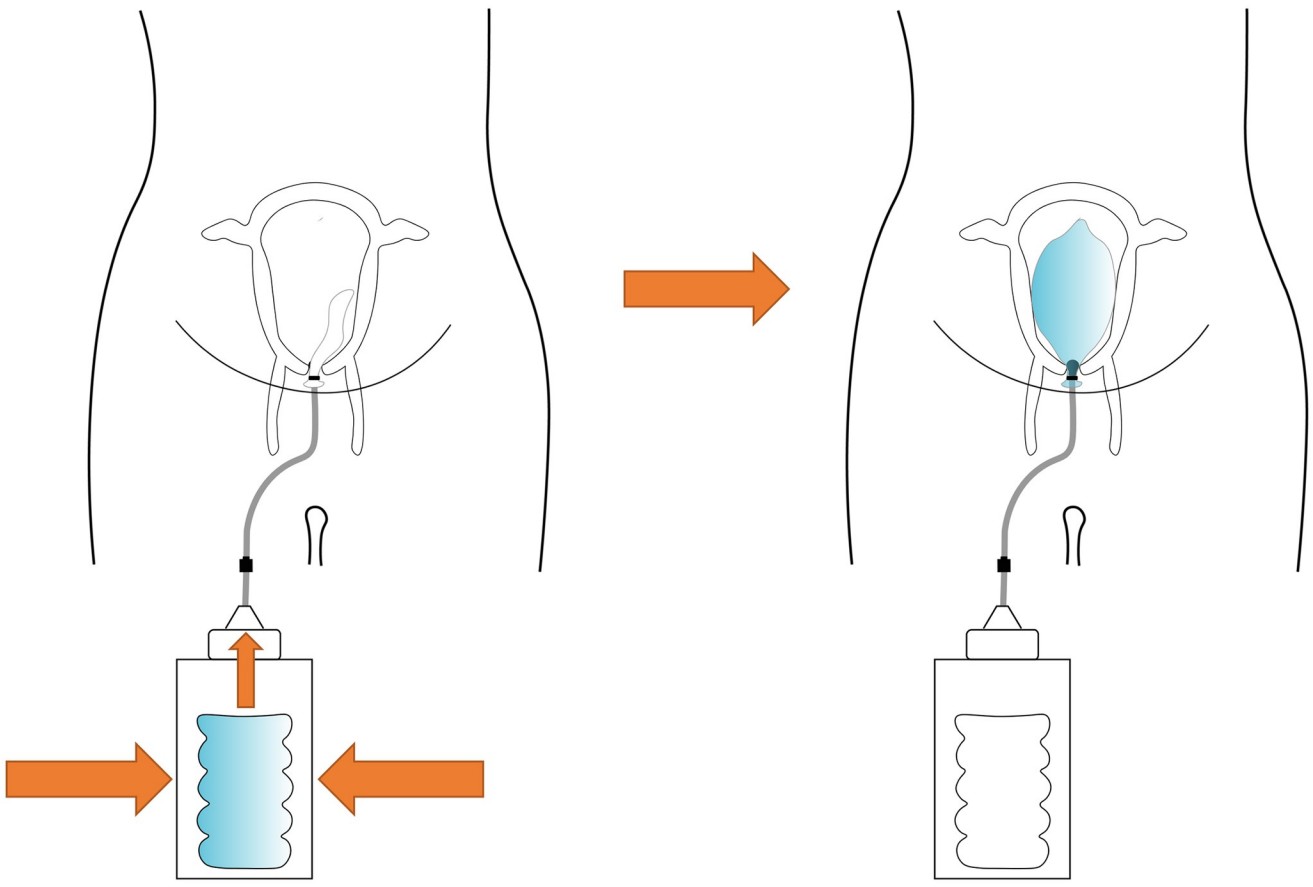

**Fig 4. The proposed mechanism of the intrauterine balloon tamponade in treating postpartum haemorrhage.** When the water reservoir is emptied into the condom balloon, the balloon becomes inflated, applying pressure inside the uterus.

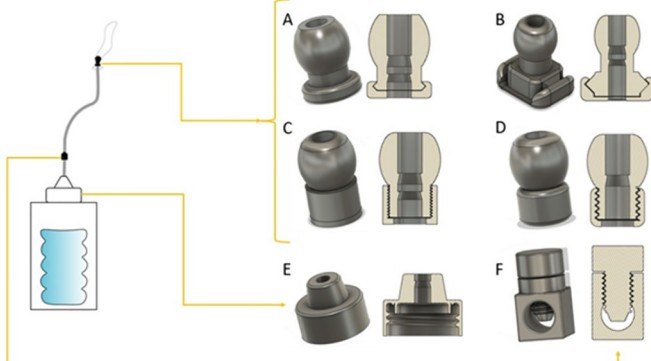

**Fig 5. The prototype system, with CAD images of parts for 3D printing: A-D valves of different design: A, rounded snap-fit valve, B, square snap-fit valve, C, fine threaded valve, D, coarse-threaded valve; E, modified bottle cap with plug and v seals, compatible with common soda bottles; F, flow stopper, a variety of globe valve preventing flow back of water down the tube.**

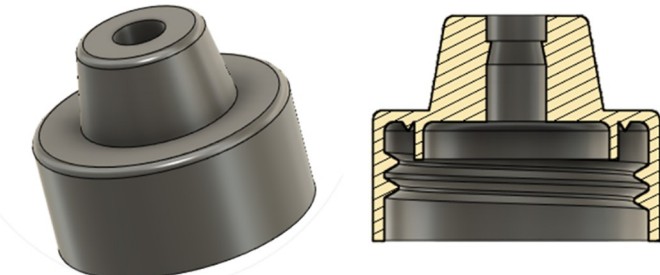

**Fig 6. The modified bottle cap with the plug and v seals.**

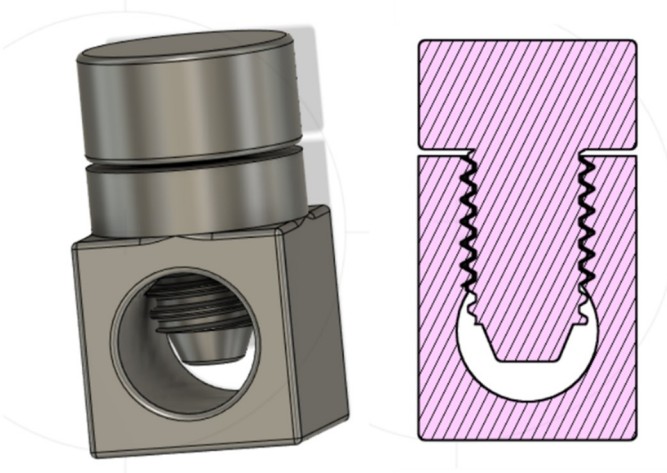

**Fig 7. The flow stopper, a variety of globe valve.**

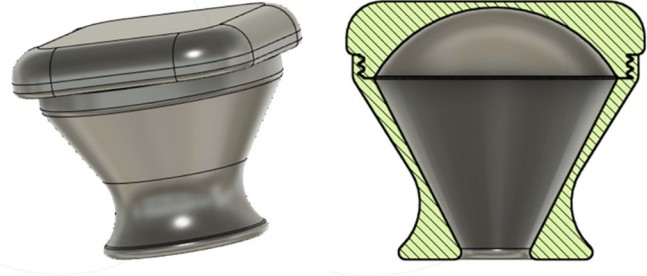

**Fig 8. The uterus model.**

protocycled into PLA filament and given new life as novel components or other 3D printed material.

## Technical validation

**Water leak.** Valve A resulted in a mean water loss of 24.75 mL/hour (1.24%/h of the infill volume) and a mean total water loss of 148.5 mL (7.43% of the infill volume) over 6 hours. The

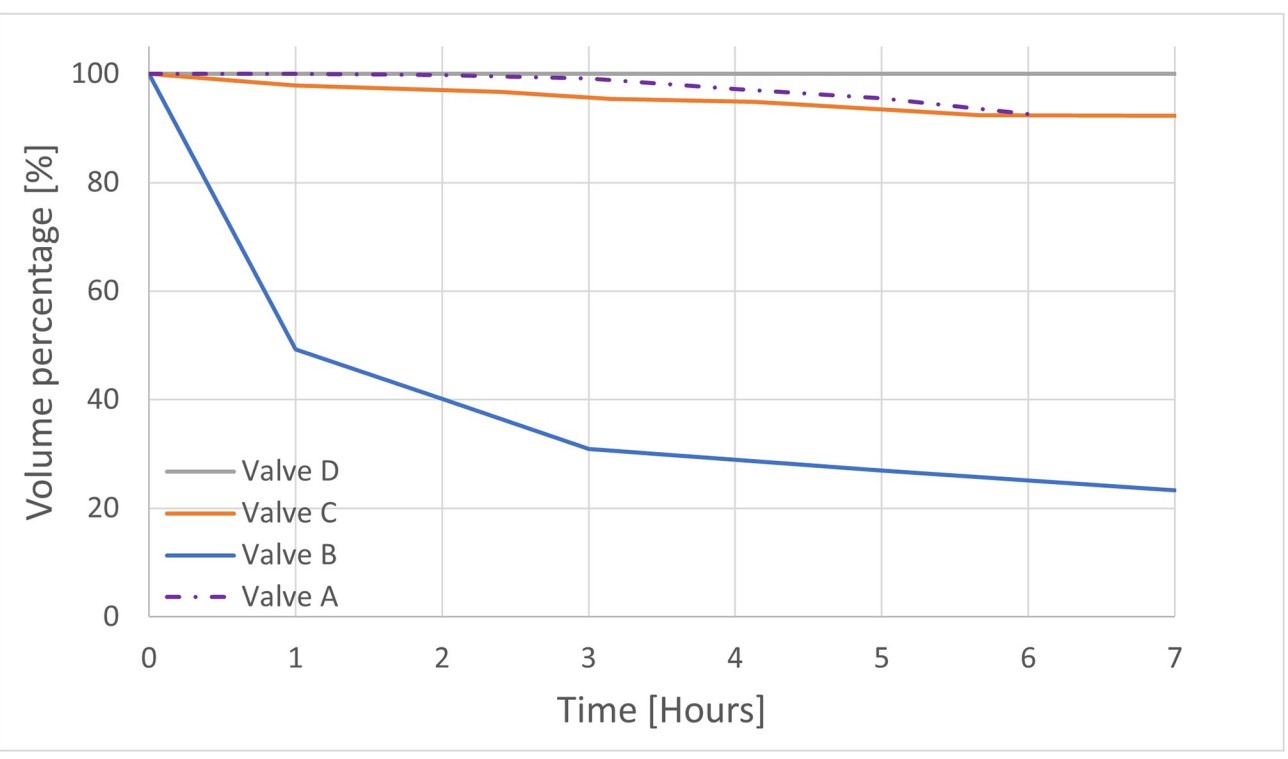

**Fig 9. The hourly water leakage for the 4 different types of valves.**

performance of Valve B, C and D, resulted in a mean water loss respectively of 53.17 mL/hour (12.93%/h of the infill volume), 5.31 mL/hour (1.29%/h of the infill volume), and 0 mL/hour, and a mean total water loss of 319 mL (77.6% of the infill volume), 31.85 mL (7.75% of the infill volume), and 0 mL over 7 hours (see Fig 9). At this point, valve D, the only valve, which did not lose water both when filled up with 400 mL or 1500 mL of water, was selected for further testing. In all the above cases, the only point of leakage was the valve connecting the condom with the silicone tube.

**Water leak in uterus model.** Two prototypes of Valve D were tested in the 3D-printed uterus model in two positions. In position 1, the uterus model was suspended so that the "cervix" opening was pointing downwards (worst case scenario), the average water loss per hour was 2.47 mL/hour (0.62%/hour of the infill volume) and the mean total water loss was 14 mL (3.51% of the infill volume). In position 2, the uterus model was laid sideways so that the longitudinal axis was almost parallel to the ground (simulating lying position), the average water loss per hour was 1.4 mL/h (0.35%/hour of the infill volume) and the mean total water loss was 7 mL (1.75% of the infill volume) over 5 hours.

**Pressure.** The water pressure circuit, constructed to monitor the pressure drop in the system resulting from the water loss, was validated against an analogic manometer (gold standard) and the resulting Bland Altman plots are shown in Fig 10. Our system had a very good percentage of agreement with the standard, apart from a 10.05-mmHg bias, which was accounted for in the measurement of pressures during the following tests. The average water pressure drop was 2.75 mmHg across the two performed experiments (0.92 mmHg per hour), and the average water leakage was 8.5 mL (2.83 mL per hour) over 3 hours. However, it is likely that most of this leakage was due to the experimenter's handling when swapping the connected

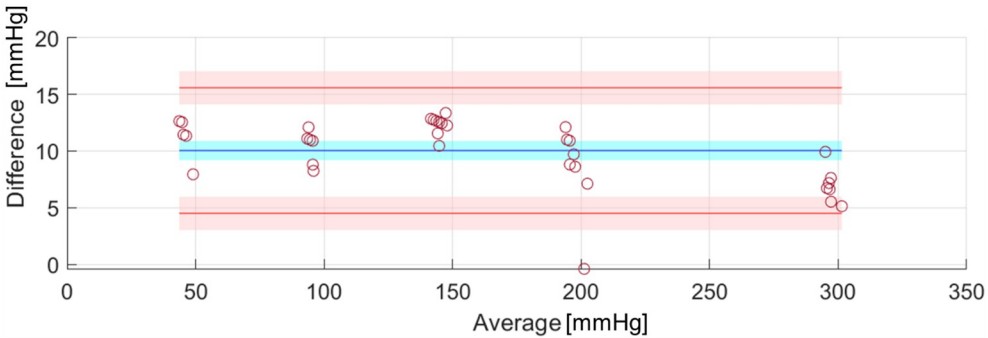

**Fig 10. Bland-Altman plot showing the 95% limits of agreement (red lines) with the 95% confidence interval (red region) and the bias (blue) with its 95% confidence interval (blue region).**

sensors (this was noticed by the experimenter, who did not see any water leakage on the tray below the system before the manual handling took place). In all the above cases, the only point of leakage was the valve connecting the condom with the silicone tube.

## Comparison with other balloons

Tables 1–4 report the results of the FMEA that was performed. It can be noticed that our device shows lower critical numbers (i.e., the product of the likelihood and the severity of a given hazard). Our device, in fact, while having comparable critical numbers in terms of mechanical, usability, and clinical risks with Bakri balloon, outperforms it in terms of cost risks and health technology management (Medium VS High risk). When compared to the condom-based tamponade, it outperforms it in terms of usability risks (Medium VS Medium High) and of health technology management risks (Medium VS Medium High). Finally, our device performance is similar (actually slightly better than) to that of BAMBI. Our device out-performs BAMBI in terms of risks related to health technology management (Medium VS Medium High).

## Discussions & conclusions

This paper presented the design and the technical validation of a 3D-printed condom-based intrauterine balloon tamponade, intended to be manufactured and used in LRSs. A frugal engineering and design ethnography perspective was adopted, in conjunction with European

**Table 1. Risk assessment for our device.** PO stands for probability of occurrence, S stands for severity, and CN stands for critical number. †denotes hypothesized usability risk, as no usability study was performed. *denotes hypothesized clinical risks, as no clinical trial was performed (or if so, results were not published).

**Our device**

| Hazard | Hazardous situation | Harm | PO | S | CN |
|---|---|---|---|---|---|
| Mechanical risks | Some components breaking/not holding the system together. | Ineffective use during PPH. If no alternative, patient death. | 1 | 4 | 4 |
| Biological risks | Some components not biocompatible. | Interaction with the human body and possible adverse reaction. | 1 | 2 | 2 |
| Usability risks | Device too difficult to install and not very usable. | Ineffective use during PPH. If no alternative, patient death. | 1† | 4 | 4† |
| Cost risks | Device and parts cost too much and cannot be afforded. | No possibility of using it during PPH events. If no alternative, patient death. | 1 | 4 | 4 |
| Health tech. management risks | Device and parts difficult to retrieve locally. | No possibility of using it during PPH events. If no alternative, patient death. | 1 | 4 | 4 |
| Clinical risks | Device not effective at stopping PPH. | Excessive blood loss, patient death. | 1* | 4 | 4* |

**Table 2. Risk assessment for Bakri balloon.** PO stands for probability of occurrence, S stands for severity, and CN stands for critical number.

| Bakri balloon | | | | | |
|---|---|---|---|---|---|
| **Hazard** | **Hazardous situation** | **Harm** | **PO** | **S** | **CN** |
| Mechanical risks | Some components breaking apart/not holding the system together. | Ineffective use during PPH. If no alternative, death of patient. | 1 | 4 | 4 |
| Biological risks | Some components not biocompatible. | Interaction with the human body and possible adverse reaction. | 1 | 2 | 2 |
| Usability risks | Device too difficult to install and not very usable. | Ineffective use during PPH. If no alternative, death of patient. | 1 | 4 | 4 |
| Cost risks | Device and parts cost too much and cannot be afforded. | No possibility of using it during PPH events. If no alternative, death of patient. | 4 | 4 | 16 |
| Health tech. management risks | Device and parts difficult to retrieve locally. | No possibility of using it during PPH events. If no alternative, death of patient. | 4 | 4 | 16 |
| Clinical risks | Device not effective at stopping PPH. | Excessive blood loss, death of patient. | 1 | 4 | 4 |

**Table 3. Risk assessment for condom-based tamponade.** PO stands for probability of occurrence, S stands for severity, and CN stands for critical number.

| Condom-based tamponade | | | | | |
|---|---|---|---|---|---|
| **Hazard** | **Hazardous situation** | **Harm** | **PO** | **S** | **CN** |
| Mechanical risks | Some components breaking apart/not holding the system together. | Ineffective use during PPH. If no alternative, death of patient. | 2 | 4 | 8 |
| Biological risks | Some components not biocompatible. | Interaction with the human body and possible adverse reaction. | 1 | 2 | 2 |
| Usability risks | Device too difficult to install and not very usable. | Ineffective use during PPH. If no alternative, death of patient. | 2 | 4 | 8 |
| Cost risks | Device and parts cost too much and cannot be afforded. | No possibility of using it during PPH events. If no alternative, death of patient. | 1 | 4 | 4 |
| Health tech. management risks | Device and parts difficult to retrieve locally. | No possibility of using it during PPH events. If no alternative, death of patient. | 2 | 4 | 8 |
| Clinical risks | Device not effective at stopping PPH. | Excessive blood loss, death of patient. | 1 | 4 | 4 |

**Table 4. Risk assessment for BAMBI device.** PO stands for probability of occurrence, S stands for severity, and CN stands for critical number. *denotes hypothesized clinical risks, as no clinical trial was performed (or if so, results were not published). ^denotes that there is not a clear costing of the device mentioned in the relevant publications. Supposedly the cost should be comparable to that of the condom-based tamponade, and less than 5 euro per kit.

| BAMBI device | | | | | |
|---|---|---|---|---|---|
| **Hazard** | **Hazardous situation** | **Harm** | **PO** | **S** | **CN** |
| Mechanical risks | Some components breaking apart/not holding the system together. | Ineffective use during PPH. If no alternative, death of patient. | 1 | 4 | 4 |
| Biological risks | Some components not biocompatible. | Interaction with the human body and possible adverse reaction. | 1 | 2 | 2 |
| Usability risks | Device too difficult to install and not very usable. | Ineffective use during PPH. If no alternative, death of patient. | 1 | 4 | 4 |
| Cost risks | Device and parts cost too much and cannot be afforded. | No possibility of using it during PPH events. If no alternative, death of patient. | 1^ | 4 | 4^ |
| Health tech. management risks | Device and parts difficult to retrieve locally. | No possibility of using it during PPH events. If no alternative, death of patient. | 2 | 4 | 8 |
| Clinical risks | Device not effective at stopping PPH. | Excessive blood loss, death of patient. | 1* | 4 | 4* |

regulations and international standards. This paper presented how contextual design principles can be taken into account and influence the final product and highlighted the results and importance of the technical validation, namely the water loss and pressures measurements within the balloon, for this particular application.

From the results, it was clear that out of the four designed valves, Valve D, i.e., the one with a coarser thread, showed the best performance. In fact, it outperformed the other valves as there was no water loss (either when filled with 500 mL or 1500 mL) over 6 hours in open air.

Moreover, when tested inside the uterus model, the mean total loss of water was negligible (i.e., 14 mL, 3.51% of the infill volume) in position 1 (the worst-case scenario) and even more negligible in position 2 (i.e., 7 mL, 1.75% of the infill volume). Nonetheless, to remedy this minimal water loss, the healthcare operator could potentially refill the balloon with an amount of water equal to the leaked one, if deemed necessary.

With regards to the pressures at play, the average pressure measured at the start of the experiments (i.e., 40.75 mmHg) was lower than those reported in [29] for the condom-catheter balloon tamponade. However, the rate of pressure loss, a factor which is of higher importance, was much lower. In fact, different studies proved that an adequate tamponade does not require high intraluminal pressures nor excessive volumes, and that adequate haemostasis, in most cases, is reached with filling volumes well below the recommended maximum [38, 39]. More specifically, Georgiou [40] demonstrated in vivo that a tamponade was efficient even when the intraluminal pressure was below the systolic blood pressure. In fact, as suggested by the author, that of exerting a pressure that is greater than the systemic arterial pressure is only one of the proposed mechanisms of this device. Other proposed mechanisms include endometrial contact, hydrostatic pressure effect on the uterine arteries, vascular compression and myometrial activity obtained through myometrial stretching.

When compared to other existing solutions for postpartum haemorrhage, the proposed solution seems to be the more versatile and appropriate for LRSs. In fact, the other solutions, some of which may have a more physiological shape (e.g., the Bakri balloon or the Ebb tamponade system), rely on parts that are either proprietary or difficult to find in LRSs (e.g., syringes, catheters, nasogastric probes, surgical wires, latex segments). Consequently, as they are single use, the local populations have to rely on the continuous supply of spare parts and are not empowered nor supported to be independent. Indeed, according to our field studies and literature search [23, 24, 37], this is one of the major bottlenecks when it comes to MDs in LRSs. As presented in the results, moreover, our FMEA highlighted a great performance of our device compare to the selected benchmarks. The closest performing one was BAMBI, which still lags behind in terms of health technology management, having to rely on parts that may be difficult to retrieve.

Overall, our solution proved to be simple, effective, and affordable. The expected cost of the system is roughly one US dollar, which more convenient compared to much higher costs of some of the similar devices mentioned before, costing well over 100 dollars [41]. The system's cost is comparable to other condom catheter devices or kits. However, we proved in other publications [37] that value is a better parameter than cost when designing MDs resilient to LRSs. In this case, as mentioned before, being able to bypass the supply chain bottleneck [42], is our added value compared to the already existing affordable solutions. The main novelty of our solution is the fact that it comprises parts that are either widely available in this kind of settings (any kind of silicon tube, condoms, and plastic bottles) or that can be manufactured locally by universities or technical centres supplied with 3D printers and protocyclers, fostering a circular economy approach, supporting local communities, and bypassing the aforementioned bottleneck of the limited supply chain in LRSs. When taking into consideration the several challenges faced by MDs in African hospitals, it seems clear that a frugal engineering and design ethnography approach, as the ones used in our case, can make the design of MDs more appropriate to LRSs working conditions [43]. Likewise, circular economy, with its 3 Rs (i.e., reduce, reuse and recycle) [44] can be one of the solutions to many of such challenges, in general and in the MD sector; in fact, it can help achieve a significant number of Sustainable Development Goals [45].

In light of this, the authors of this paper propose that a circular economy approach is feasible with our intrauterine balloon tamponade. Specifically, a protocycler (https://redetec.com/),

i.e., an all-in-one recycling system for 3D printers, could be used to recycle, reuse and probably sterilise the 3D-printed parts of our intrauterine balloon tamponade. Moreover, as previously stated, the efficacy of condom-catheters intrauterine balloons was proven by different studies and the risk of prolapse due to the non-physiological shape can be mitigated with different techniques (using forceps, gauzes and/or sutures) [46–48]. Another advantage of our system is that the high temperatures used by the 3D printers and the protocycler (over 180˚C) are enough to sterilize the instruments. Rankin et al. [49] demonstrated that the clean catch filament freshly extruded by a 3D printer was completely sterile. Thus, if the print takes place on a clean print bed and in a clean environment, the final product will most likely be sterile, as proved by Kondor et al. [50].

This study has potential limitations. In fact, the uterus model that was designed based on that presented in [29] is only a first approximation model. The "geometrical" shapes, and mainly the properties of the materials used, do not reflect the actual properties of the uterus. Hence, the results of the tests performed within this model may change if superior and more realistic models were used. Nonetheless, it is likely that, since a more realistic model would be made with a more flexible and adaptable material, compared to PLA, the pressures at play and the consequent leaks may be even lower. Creating a very realistic uterus model was out of the scope of this study. Future research may want to focus on this aspect and create a uterus model, based on real CT images of postpartum uteri and on the use of polymerised rubber, that could also be sensorised to monitor relevant variables. Moreover, it is worth recalling that this is a design concept project and clinical studies would be needed to further validate the device. Further work towards meeting essential requirements for complete compliance with MDR 2017/745 and its harmonised standards would be needed to progress to higher technology readiness levels. In terms of limitations of the FMEA analysis, in some cases, it was necessary to estimate the probability of occurrence, because either that dimension had not yet been tested with a trial or because it had not been reported by the other devices.

This paper presented the design and technical validation of a 3D-printed condom intrauterine balloon tamponade, suitable for use in LRSs. The performance of this device, as confirmed by the validation, is promising. The intrauterine balloon tamponade, in fact, proved to be able to hold large volumes of liquid and maintain the pressure stable for several hours, as per postpartum haemorrhage treatment guidelines. In addition, the system would be suitable to be manufactured and recycled locally, making it a promising option in combatting the devastating impact of postpartum haemorrhage in LRSs. Overall, this research is grounded on an important ethical substratum, fosters the enjoyment of human rights and aims at their improvement, especially for women.

## Supporting information

**S1 Fig.**
(TIF)

**S1 File.**
(XLSX)

## Acknowledgments

The authors gratefully acknowledge Dr Selene Tognarelli and Prof Arianna Menciassi (Sant'Anna School of Advanced Studies, Pisa, Italy)) for their support and precious feedback. The authors gratefully acknowledge Katy Stokes for proofreading the article and supporting the creation of some of the figures. The authors gratefully acknowledge Stefano Canavero (MSc in

Integrated Product Design at the Polytechnic University of Milan) for his support on the graphical abstract.

## Author Contributions

**Conceptualization:** Alessia Maccaro, Ernesto Iadanza, Leandro Pecchia.

**Data curation:** Davide Piaggio, Scott Hyland.

**Formal analysis:** Davide Piaggio, Scott Hyland.

**Funding acquisition:** Alessia Maccaro, Leandro Pecchia.

**Investigation:** Davide Piaggio, Scott Hyland, Leandro Pecchia.

**Methodology:** Davide Piaggio, Scott Hyland, Leandro Pecchia.

**Project administration:** Leandro Pecchia.

**Resources:** Leandro Pecchia.

**Supervision:** Ernesto Iadanza, Leandro Pecchia.

**Visualization:** Davide Piaggio, Scott Hyland, Ernesto Iadanza.

**Writing – original draft:** Davide Piaggio, Scott Hyland, Alessia Maccaro, Ernesto Iadanza, Leandro Pecchia.

**Writing – review & editing:** Davide Piaggio, Scott Hyland, Alessia Maccaro, Ernesto Iadanza, Leandro Pecchia.

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
