## [Decision Letter · Decision Letter 0]

3 May 2023

PONE-D-23-08013A 3D-printed condom intrauterine balloon tamponadePLOS ONE

Dear Dr. Piaggio,

Thank you for submitting your manuscript to PLOS ONE. After careful consideration, we feel that it has merit but does not fully meet PLOS ONE’s publication criteria as it currently stands. Therefore, we invite you to submit a revised version of the manuscript that addresses the points raised during the review process.

Dear Dr. Piaggio,

Thank you for submitting your manuscript and thank you for your patience. We have been able to receive two reviews, however, we feel that your manuscript requires at least one review by a researcher in the biomedical engineering field, which has taken longer than we had anticipated. For now, please respond to the comments by the first reviewers and return the manuscript with responses to their comments as we work on securing a review from a biomedical engineering researcher, whose comments might come at the next round of the review process. Please submit your revised manuscript by Jun 17 2023 11:59PM. If you will need more time than this to complete your revisions, please reply to this message or contact the journal office at plosone@plos.org. Please include the following items when submitting your revised manuscript:A rebuttal letter that responds to each point raised by the academic editor and reviewer(s). You should upload this letter as a separate file labeled 'Response to Reviewers'.A marked-up copy of your manuscript that highlights changes made to the original version. You should upload this as a separate file labeled 'Revised Manuscript with Track Changes'.An unmarked version of your revised paper without tracked changes. You should upload this as a separate file labeled 'Manuscript'.If applicable, we recommend that you deposit your laboratory protocols in protocols.io to enhance the reproducibility of your results. Protocols.io assigns your protocol its own identifier (DOI) so that it can be cited independently in the future. For instructions see: https://journals.plos.org/plosone/s/submission-guidelines#loc-laboratory-protocols. Additionally, PLOS ONE offers an option for publishing peer-reviewed Lab Protocol articles, which describe protocols hosted on protocols.io. Read more information on sharing protocols at https://plos.org/protocols?utm_medium=editorial-email&utm_source=authorletters&utm_campaign=protocols.

We look forward to receiving your revised manuscript.

Kind regards,

Daniel Semakula, M.D. MPH, PhD

Academic Editor

PLOS ONE

Journal Requirements:

Reviewers' comments:

Reviewer's Responses to Questions

**Comments to the Author**

1. Is the manuscript technically sound, and do the data support the conclusions?

Reviewer #1: Partly

Reviewer #2: Yes

2. Has the statistical analysis been performed appropriately and rigorously? 

Reviewer #1: I Don't Know

Reviewer #2: N/A

3. Have the authors made all data underlying the findings in their manuscript fully available?

Reviewer #1: No

Reviewer #2: Yes

4. Is the manuscript presented in an intelligible fashion and written in standard English?

Reviewer #1: No

Reviewer #2: Yes

5. Review Comments to the Author

Reviewer #1: The paper is addressing an important intervention component to manage PPH esp in low resource setting. However the technical aspects are not clear and comparison with available standard methods is not explicitly described. The contents of the UBT need to be described in detail what material, size, applicability and how this use is supposed to be different than others.

I do not agree with the advantage of being able to sterilise this UBT as there are multiple low cost dispensable UBTs available. Hence its not clear to the reader how this UBT is better in either the contents, material used, ease of use, capacity of the device to hold the fluid, duration that could be kept in the uterus. The figure of inflated balloon also does not visually satisfy that it takes shape of the uterus such that its entire inner surface gets pressure from the inflated balloon

I suggest the paper be reviewed by biomedical engineer expert to understand the nitty-gritties. Every device has to meet an international quality standard (ISO) . There is no mention how this device falls within those standards

Authors must also be aware that 3 D printers may not be easily available in Low resource settings for which this device is being developed.

Reviewer #2: This is very important development that should tested with real PPH cases, where the other management options like uterotonics are also exacting pressure on the uterus. Few comments for the authors to clarify

1. The abstract results are written as conclusion of the prototyping but not the results. Please state the results

2. Lines 2 and 3: Define the abbreviation LRSs and LMIC since they appear for the first time. There are several abbreviations that must be defined other than assuming that the readers know what it is.

3. You will a reference for this statement “Postpartum haemorrhage is defined as heavy bleeding after birth and it can be classed as “primary” if there is a 500-ml blood loss (>2 litres in case of severe haemorrhage) within the first 24 hours after birth, and as “secondary” if there is abnormal or heavy vaginal bleeding between 24 hours and 12 weeks after birth.”

4. Lines 18-20, the authors mention that the uterine balloon is among the first line management option, they need to check the citation well. It is for refractory management of PPH.

5. In line 82-83, The parts that underwent 3D printing were: 1) A valve 2) A flow-stopper 3) A modified bottle cap 4) A uterus model. What about the condom uterine balloon. It is not clear how this is different from ordinary condom

6. With several parts making the system, what were the different points of leakage of water.

7. Although the circular economic model is mentioned in the introduction and the discussion, the authors have not demonstrated how it applied in the methods and results. Which parts are re-usable.

8. What is the likely cost of the device

6. PLOS authors have the option to publish the peer review history of their article (what does this mean?). If published, this will include your full peer review and any attached files.

Reviewer #1: No

Reviewer #2: **Yes: **Sam Ononge

---

## [Author Response · Author response to Decision Letter 0]

28 Jun 2023

Thank you for your precious feedback. Please see detailed response in the attached word.

---

## [Decision Letter · Decision Letter 1]

23 Nov 2023

PONE-D-23-08013R1A 3D-printed condom intrauterine balloon tamponadePLOS ONE

Dear Dr. Piaggio,

Thank you for submitting your manuscript to PLOS ONE. After careful consideration, we feel that it has merit but does not fully meet PLOS ONE’s publication criteria as it currently stands. Therefore, we invite you to submit a revised version of the manuscript that addresses the points raised during the review process. Please submit your revised manuscript by Jan 07 2024 11:59PM. If you will need more time than this to complete your revisions, please reply to this message or contact the journal office at plosone@plos.org. Please include the following items when submitting your revised manuscript:A rebuttal letter that responds to each point raised by the academic editor and reviewer(s). You should upload this letter as a separate file labeled 'Response to Reviewers'.A marked-up copy of your manuscript that highlights changes made to the original version. You should upload this as a separate file labeled 'Revised Manuscript with Track Changes'.An unmarked version of your revised paper without tracked changes. You should upload this as a separate file labeled 'Manuscript'.

We look forward to receiving your revised manuscript.

Kind regards,

Hugh Cowley

Staff Editor

PLOS ONE

**Additional Editor Comments:**

Your manuscript has been evaluated by two reviewers: one from the previous round (Reviewer 1) and one new reviewer with expertise in biomedical engineering (Reviewer 3). Their comments are appended below.

Reviewer 1 is satisfied with the changes you have made. Reviewer 3 recognizes the changes, but has stated that clinical experiments and a comparison with standard methods are necessary to support the conclusions of this study. Recognizing that this work was intended as a pre-clinical investigation, clinical validation will be a necessary step, but reasonable to consider outside the scope of the current study. However, demonstration of performance against existing comparable balloon tamponades would help validate the claim that this newly proposed concept has comparable (or at least practical given the other stated advantages) functionality. Please also ensure that any claims of potential efficacy are appropriately qualified with statements clarifying that no clinical experiments have been carried out.

Reviewers' comments:

Reviewer's Responses to Questions

**Comments to the Author**

1. If the authors have adequately addressed your comments raised in a previous round of review and you feel that this manuscript is now acceptable for publication, you may indicate that here to bypass the “Comments to the Author” section, enter your conflict of interest statement in the “Confidential to Editor” section, and submit your "Accept" recommendation.

Reviewer #1: All comments have been addressed

Reviewer #3: All comments have been addressed

2. Is the manuscript technically sound, and do the data support the conclusions?

Reviewer #1: Yes

Reviewer #3: Partly

3. Has the statistical analysis been performed appropriately and rigorously? 

Reviewer #1: N/A

Reviewer #3: No

4. Have the authors made all data underlying the findings in their manuscript fully available?

Reviewer #1: Yes

Reviewer #3: Yes

5. Is the manuscript presented in an intelligible fashion and written in standard English?

Reviewer #1: Yes

Reviewer #3: Yes

6. Review Comments to the Author

Reviewer #1: (No Response)

Reviewer #3: I really appreciate all your hard work to write this article. however, it still needs more efforts to finalize it. A set of clinical experiments is needed to validate the proposed catheter. Also, a comparison with one of the standard methods is highly required to support the concluded results.

best regards,

7. PLOS authors have the option to publish the peer review history of their article (what does this mean?). If published, this will include your full peer review and any attached files.

Reviewer #1: No

Reviewer #3: No

---

## [Decision Letter · Decision Letter 2]

11 Mar 2024

PONE-D-23-08013R2A 3D-printed condom intrauterine balloon tamponade: design, prototyping, and technical validationPLOS ONE

Dear Dr. Piaggio,

Thank you for submitting your manuscript to PLOS ONE. After careful consideration, we feel that it has merit but does not fully meet PLOS ONE’s publication criteria as it currently stands. Therefore, we invite you to submit a revised version of the manuscript that addresses the points raised during the review process.

Dear authors,

please correct the manuscript by separating the single paragraph of the discussion into 3 or 4 and leave it in accordance with the following reviewers' recommendations: "this version of the manuscript has more typos than the previous one (e.g. periods in titles of some subsections of the " Results" section and the "Discussions and Conclusions" section consists of only one long paragraph). Also, the reference section needs to be revised carefully (e.g. ref 23 and 24 are repeated in 37 and 38).", and "the quality of Figure 1 must be improved and the Figure 2 showing the complete prototype, including the used pump."

We look forward to receiving your revised manuscript.

Kind regards,

Ricardo Ney Oliveira Cobucci, Ph.D

Academic Editor

PLOS ONE

Journal Requirements:

Additional Editor Comments:

Dear authors,

please correct the manuscript by separating the single paragraph of the discussion into 3 or 4 and leave it in accordance with the following reviewers' recommendations: "this version of the manuscript has more typos than the previous one (e.g. periods in titles of some subsections of the " Results" section and the "Discussions and Conclusions" section consists of only one long paragraph). Also, the reference section needs to be revised carefully (e.g. ref 23 and 24 are repeated in 37 and 38).", and "the quality of Figure 1 must be improved and the Figure 2 showing the complete prototype, including the used pump."

Reviewers' comments:

Reviewer's Responses to Questions

**Comments to the Author**

1. If the authors have adequately addressed your comments raised in a previous round of review and you feel that this manuscript is now acceptable for publication, you may indicate that here to bypass the “Comments to the Author” section, enter your conflict of interest statement in the “Confidential to Editor” section, and submit your "Accept" recommendation.

Reviewer #1: All comments have been addressed

Reviewer #3: (No Response)

Reviewer #4: All comments have been addressed

Reviewer #5: All comments have been addressed

2. Is the manuscript technically sound, and do the data support the conclusions?

Reviewer #1: Yes

Reviewer #3: No

Reviewer #4: Yes

Reviewer #5: Yes

3. Has the statistical analysis been performed appropriately and rigorously? 

Reviewer #1: Yes

Reviewer #3: No

Reviewer #4: I Don't Know

Reviewer #5: Yes

4. Have the authors made all data underlying the findings in their manuscript fully available?

Reviewer #1: (No Response)

Reviewer #3: Yes

Reviewer #4: Yes

Reviewer #5: Yes

5. Is the manuscript presented in an intelligible fashion and written in standard English?

Reviewer #1: Yes

Reviewer #3: No

Reviewer #4: Yes

Reviewer #5: Yes

6. Review Comments to the Author

Reviewer #1: (No Response)

Reviewer #3: I reject this paper mainly for the lack of solid statistical comparisons between the proposed system and other systems which are already in the market. Although, the authors showed some comparisons in separate tables, I am still not satisfied by the point of comparisons or the qualitative results used. Also, the comments from the previous round were not completely addressed. The manuscript is still not acceptable for publication and this version of the manuscript has more typos than the previous one (e.g. periods in titles of some subsections of the "Results" section and the "Discussions and Conclusions" section consists of only one long paragraph). Also, the reference section needs to be revised carefully (e.g. ref 23 and 24 are repeated in 37 and 38).

Reviewer #4: (No Response)

Reviewer #5: The paper presents a proposal for a hemorrhages intrauterine tamponade prototype that may be occur in the postpartum. Although this technique is not new, the paper has a target audience in low-resource settings, propose the use of recycled material and presents innovation of 3D printing use for some parts of the prototype, which facilitates the dissemination to the proposed device.

I noted that the requested revisions were met.

The paper target all the requirements to the publication and I believe it has good potential to encourage more research in this study area.

In my point of view, the paper should be accepted for publication.

My suggestion is the quality of Figure 1 to be improve and that the Figure 2 showing the complete prototype, including the used pump.

7. PLOS authors have the option to publish the peer review history of their article (what does this mean?). If published, this will include your full peer review and any attached files.

Reviewer #1: No

Reviewer #3: No

Reviewer #4: **Yes: **Itamir de Morais Barroca Filho

Reviewer #5: **Yes: **Jan Erik Mont Gomery Pinto

---

## [Editor Report · Decision Letter 3]

2 May 2024

A 3D-printed condom intrauterine balloon tamponade: design, prototyping, and technical validation

PONE-D-23-08013R3

Dear Dr. Piaggio,

We’re pleased to inform you that your manuscript has been judged scientifically suitable for publication and will be formally accepted for publication once it meets all outstanding technical requirements.

Kind regards,

Ricardo Ney Oliveira Cobucci, Ph.D

Academic Editor

PLOS ONE
---

## [Editor Report · Acceptance letter]

14 May 2024

PONE-D-23-08013R3 

PLOS ONE

Dear Dr. Piaggio, 

I'm pleased to inform you that your manuscript has been deemed suitable for publication in PLOS ONE. Congratulations! Your manuscript is now being handed over to our production team.

Kind regards, 

on behalf of

PROFESSOR Ricardo Ney Oliveira Cobucci 

Academic Editor

PLOS ONE